# Mycobacteriophages as Potential Therapeutic Agents against Drug-Resistant Tuberculosis

**DOI:** 10.3390/ijms22020735

**Published:** 2021-01-13

**Authors:** Anna Allué-Guardia, Rajagopalan Saranathan, John Chan, Jordi B. Torrelles

**Affiliations:** 1Population Health Program, Tuberculosis Group, Texas Biomedical Research Institute, San Antonio, TX 78227, USA; 2Department of Microbiology and Immunology, Albert Einstein College of Medicine, New York, NY 10461, USA; saranathan.rajagopalan@einsteinmed.org (R.S.); john.chan@einsteinmed.org (J.C.)

**Keywords:** *Mycobacterium tuberculosis*, drug-resistance, mycobacteriophages, phage therapy, lung mucosa

## Abstract

The current emergence of multi-, extensively-, extremely-, and total-drug resistant strains of *Mycobacterium tuberculosis* poses a major health, social, and economic threat, and stresses the need to develop new therapeutic strategies. The notion of phage therapy against bacteria has been around for more than a century and, although its implementation was abandoned after the introduction of drugs, it is now making a comeback and gaining renewed interest in Western medicine as an alternative to treat drug-resistant pathogens. Mycobacteriophages are genetically diverse viruses that specifically infect mycobacterial hosts, including members of the *M. tuberculosis* complex. This review describes general features of mycobacteriophages and their mechanisms of killing *M. tuberculosis*, as well as their advantages and limitations as therapeutic and prophylactic agents against drug-resistant *M. tuberculosis* strains. This review also discusses the role of human lung micro-environments in shaping the availability of mycobacteriophage receptors on the *M. tuberculosis* cell envelope surface, the risk of potential development of bacterial resistance to mycobacteriophages, and the interactions with the mammalian host immune system. Finally, it summarizes the knowledge gaps and defines key questions to be addressed regarding the clinical application of phage therapy for the treatment of drug-resistant tuberculosis.

## 1. Introduction

Tuberculosis (TB) is the leading cause of mortality worldwide due to a single infectious disease, with an estimate of ~1.4 million attributed deaths in 2019 [1]. It is caused by the airborne pathogen *Mycobacterium tuberculosis* (*M. tuberculosis*), which upon inhalation is deposited in the lung alveolar space. In that environment, *M. tuberculosis* comes in close contact with soluble components of the human alveolar lining fluid (ALF) prior to and following its encounter with alveolar resident and compartment cells [2,3]. After infection of alveolar cells such as alveolar macrophages (AMs), a host immune response is mounted to control the infection driving clearance, active TB disease or, in the majority of the cases, latent *M. tuberculosis* infection, which is controlled and maintained in a complex cellular structure called granuloma.

When active TB disease is developed, the standard treatment consists of a 6-month regimen using a combination of four first-line drugs (isoniazid, rifampicin, ethambutol, and pyrazinamide) that, although very efficient, can promote the emergence of resistance in the absence of sufficient healthcare infrastructure and patient poor adherence to therapy [4]. As a direct consequence, in the past two decades, multi (MDR), extensively (XDR), extremely (XXDR), and total (TDR) drug-resistant *M. tuberculosis* strains have emerged worldwide as a threat to public health and TB control, especially in TB endemic areas [5,6,7]. MDR strains are defined as resistant to at least the two first-line drugs isoniazid and rifampicin, whereas XDR, first reported in 2006 in South Africa, are resistant to isoniazid and rifampicin plus any fluoroquinolone, and at least one of the three injectable second-line drugs (amikacin, kanamycin, or capreomycin) [8]. Treatment of drug-resistant *M. tuberculosis* strains is lengthy (up to two years) and costly (worldwide the median cost per person treated for TB in 2019 was US$ 860 for drug-susceptible TB vs. US$ 5,659 for MDR-TB) [1]. The latest treatment outcome data in the WHO 2020 TB report show success rates of 85% for susceptible TB, while only 57% for MDR-TB [1]. XXDR *M. tuberculosis* strains are resistant to all first and second-line drugs [9]. The boundaries between XXDR and TDR have not been fully defined [10], and even though TDR is a term that has not been recognized by the WHO [11], *M. tuberculosis* strains resistant to all tested antibiotics, including some that are still in the discovery pipeline, have been identified in India, Iran, Italy, and South Africa, and designated as “Totally drug-resistant” [6,7]. This makes XXDR and TDR *M. tuberculosis* strains virtually incurable [12].

With the global burden of MDR-TB increasing at an annual rate of >20% in the past few years, it is estimated that drug-resistant TB will kill 75 million people and cost the global economy $16.7 trillion over the next 35 years, stressing the need to develop new drugs and/or alternative anti-TB therapies [13,14]. Currently, 22 drugs for the treatment of drug-susceptible TB, MDR-TB or latent TB are in Phase I, II or III trials, including 13 new compounds and six repurposed drugs, among others [1]. Most of the current treatments target the *M. tuberculosis* cell envelope, a complex structure comprised mainly of carbohydrates and lipids. However, little is known about the cell envelope composition of drug-resistant *M. tuberculosis* strains and their adaptation to the lung environment, which makes it challenging to develop new effective drug regimens [15]. Such strains bring us back to the pre-antibiotic era and emphasize the urgent need to develop alternative strategies to combat the disease.

The idea of using bacteriophages (virus that infect bacteria) to treat infectious diseases was first introduced at the beginning of the 20th century, shortly after Felix d’Herelle described the appearance of small, clear areas (later called plaques) in agar cultures of *Shigella* strains incubated with fecal filtrates from dysentery patients, which he proposed were caused by virus capable of “eating” bacteria or bacteriophages (derived from “bacteria” and the Greek φαγεῖν or *phagein*, “to eat”) [16,17]. During the 1920′s, 30′s and 40′s bacteriophages were commercially produced in several countries for the treatment or prophylaxis of several diseases such as dysentery, cholera, skin lesions or respiratory tract infections, among others [18,19,20]. However, at the time, the efficacy and safety of commercial phage preparations was questionable, not always reproducible, and the lack of detailed information on phage titers, host range, control groups, and experimental design in clinical trials promoted a growing skepticism towards phage therapy in the scientific community [17,21]. As a result, and with the introduction of penicillin and other antibiotics in the 1940′s, the use of phage therapy was abandoned in Western countries. However not in the former Soviet Union, where intensive research is still being conducted at the Eliava Institute of Bacteriophage in Tbilisi (Georgia), and phage cocktails are currently approved for the treatment and/or prophylaxis of several diseases [22,23,24,25,26].

Even though the notion of phage therapy has been around for nearly a century, it is still considered an experimental treatment in Western countries and has not been approved for human use yet, in part due to non-existing phage therapy regulations and the lack of safe and well-described phage preparations [27,28,29]. In addition, the fact that phage therapy is conceived as personalized medicine presents extra challenges in the regulatory pipeline. In the US, the use of phages is only approved for the decontamination of food, plant-based and livestock products, or surfaces [21,30,31,32,33,34]. However, the Food and Drug Administration (FDA) has recently approved the new Center for Innovative Phage Applications and Therapeutics (IPATH) to use phage therapy as an experimental treatment through the Emergency Investigational New Drug scheme [28], focused on cases of serious or life-threatening conditions, when no satisfactory alternative therapy is available and the patient cannot receive treatment through any existing approved clinical trial. It is expected that initiatives like this will generate supporting data for phage therapy, gaining the trust of clinicians and researchers, for implementation in the near future. With the worldwide emergence of multidrug-resistant strains for most bacterial diseases, including TB, antibiotics are becoming ineffective, with only a few new drugs in the antibiotic pipeline expected to be available in the next few years [35]. Thanks to vast advancements and a better understanding on phage biology and genomics over these past decades, as well as the development of new molecular biology tools for the study of bacteriophages and their interactions within the human body, we have now the framework to conduct exhaustive phage therapy studies, which opens the avenue for the potential use of phages as an alternative therapy when antibiotic treatment is not possible. As a consequence, these past few years have seen a renewed interest in the use of bacteriophages to treat multidrug-resistant bacteria, as well as chronic and persistent infections [21,36,37,38,39,40], with some experimental studies being conducted in the TB field. In this context, phages were successfully used to treat a patient with disseminated multidrug-resistant *Mycobacterium abscessus* [37].

This review will focus on what we know about mycobacteriophages and their applications to kill *M. tuberculosis*, as well as recently developed tools for bacteriophage engineering, with a particular emphasis in the current and future status on the use of mycobacteriophages to treat drug-resistant TB.

## 2. Mycobacteriophages

### 2.1. General Features

Mycobacteriophages are virus that specifically infect mycobacterial hosts, first isolated in the 1940s [41]. Upon isolation, one of their initial applications was their use for the typing of clinical *Mycobacterium* spp. (Table 1). The fact that some phages have narrowed host range specificities, e.g., infecting particular mycobacterial species and even strains, allowed the use of defined mycobacteriophage sets with pre-determined host specificities to identify newly isolated mycobacterial strains [42,43]. However, with the development of new molecular tools, phage typing was replaced by DNA-based methods [43]. The following years, several characterization studies were conducted [44,45,46,47,48], which would establish the framework for the subsequent studies on mycobacteriophage applications in *M. tuberculosis* genetics [49,50]. In the 1980′s, the description of shuttle phasmids as DNA transporters between *E. coli* plasmids and mycobacteriophages paved the way for recombinant DNA research in mycobacteria [51], and contributed to the subsequent development of mycobacteriophage-based methods for the genetic manipulation of mycobacterial DNA. Since then, mycobacteriophages have been the key in development of genetic tools to engineer mycobacteria [52], including transduction of chromosomal markers and transposons [53], introduction of point mutations [54], the construction of *M. tuberculosis* deletion mutants [55,56,57], and/or the introduction of immunity-based selection markers in mycobacterial hosts [58] (Table 1). Reporter mycobacteriophages have been also constructed through both “recombineering” (genetic engineering by phage recombination) and shuttle phasmid approaches, and used for TB diagnostic applications to assess active disease by determining the presence of viable *M. tuberculosis* bacilli, drug susceptibility or treatment efficacy [59,60,61] (Table 1). In this regard, luciferase reporter phages (LRPs) carrying the firefly luciferase gene (*fflux*) were first engineered in the 1990s [61] using different mycobacteriophages such as L5, D29, Che12 or TM4 [62,63], and have been improved since then by using the nanoluciferase (Nluc) reporter enzyme [64]. Further, fluoromycobacteriophages carrying the green fluorescent protein gene (*gfp*) cassette were developed in 2009 [65] and used as a fast, highly specific and sensitive method for the detection of *M. tuberculosis* drug-resistant populations by measuring fluorescence emitted by GFP-expressing bacilli after phage infection [66,67,68,69]. Later, a dual-reporter mycobacteriophage (Φ^2^DRMs) was constructed by expressing GFP (*mVenus*) constitutively after *M. tuberculosis* infection and tdTomato (red fluorescent protein) through the dnaK promoter, which is specifically upregulated in persisters, in order to identify and quantitate *M. tuberculosis* persister cells after antibiotic treatment [70]. Mycobacteriophages could also be used as potential prophylactic and/or therapeutic agents (Table 1), applications that will be further discussed later in this review.

With the advancement of new sequencing technologies during late 20th century, the first complete genome of mycobacteriophages were sequenced [50], providing better insight into their genetics and diversity. Till date, 11,468 mycobacteriophages have been isolated and 1971 mycobacteriophage genomes sequenced [71,72] (as of 5 January 2021). These phages have been mainly isolated and characterized from the environment, using non-pathogenic and fast-growing mycobacterial hosts such as *M. smegmatis* [73,74].

All described mycobacteriophages to date belong to the *Caudoviridae* type, characterized by an icosahedral head with double-stranded DNA (dsDNA) and a tail [75,76], and specifically to the *Myoviridae* (with contractile tail) and *Siphoviridae* (non-contractile tail) families, the later containing >90% of the mycobacteriophages described [77]. As for their genomic diversity, mycobacteriophages constitute architectural mosaics, evidenced by the presence of segments with distinctive evolutionary histories acquired or lost through different recombination events [78,79,80]. Still, most mycobacteriophages share a similar overall genome architecture, with structural and assembly genes (~50% of the genome) at the left end of the genome [81], followed by a bacterial lysis system and genes responsible for the lytic/lysogenic life cycles, including integration or partitioning systems [82]. Other non-structural genes are tRNA and tmRNA, potentially involved in gene expression [80,83], mobile genetic elements (MGEs) [84,85], and several other genes of unknown function [55]. The latter do not seem to be required directly for phage replication, but instead might modulate bacteria-phage and/or phage-phage dynamic interactions [86,87]. Indeed, temperate mycobacteriophages, which can either undergo lytic or lysogenic cycle (Figure 1A), play important roles in these interactions both during lysogeny by prophage-mediated defense systems against the attack of other phages, and by superinfection exclusion during lytic replication, reviewed in detail elsewhere [77,88,89,90].

Based on their overall nucleotide similarity, mycobacteriophages are grouped in different clusters and sub-clusters, where phages from the same cluster share >50% of their genomic DNA or >35% of their genes [77]. Currently, there are 29 defined mycobacteriophage clusters [A to Z, AA, AB, and AC], being A the largest cluster; and 16 singletons, which are not genetically related to any of the other clusters and constitute separate groups by themselves. Twelve clusters are further divided into sub-clusters: A1 to A14 and A16 to A20 (A15 phages only infect *Gordonia* strains), B1 to B13, C1 and C2, D1 and D2, F1 to F5, G1 to G5, H1 and H2, I1 and I2, K1 to K7, L1 to L4, M1 to M3, and P1 to P6. Sequenced mycobacteriophages have been mostly isolated in *M. smegmatis*, although some of them are also reported to infect other mycobacterial species such as *M. abscessus*, *M. aichiense*, *M. aurum*, *M. avium*, *M. phlei*, and *M. tuberculosis*; including 17 different strains among these species [72]. Phages are constantly assigned to these groups based on sequence similarity and clusters and sub-clusters created as new phage sequences are being discovered [91,92,93]. These mycobacteriophage clusters are based on sequence similarity, indicative of their mosaic architecture, which does not necessarily correlate with their biological and/or evolutionary significance.

### 2.2. Mode of Action against Mycobacterial Hosts

As phages that infect mycobacterial hosts, mycobacteriophages are equipped with the machinery to efficiently lyse mycobacteria. That includes encoding for lysins or endolysins (LysA), enzymes with the ability to cleave the peptidoglycan (PG) layer present in the mycobacterial cell envelope (Figure 1B). All mycobacteriophages, including the temperate ones, encode proteins that act as endolysins. This is the case of phage Che12, capable of infecting and lysogenizing *M. tuberculosis* [94]. These endolysins seem to be pretty diverse and modular, but most of them are composed of three domains: a C-terminal domain associated with binding to specific substrates in the bacterial cell envelope, and two catalytic domains (central and N-terminal domains) predicted to target most of the peptidoglycan bonds [95,96,97,98]. Further, most mycobacteriophages rely on endolysin-holin systems to degrade the PG layer. While endolysins target the integrity of the *M. tuberculosis* cell envelope, holins are membrane proteins that help translocate endolysins to reach their target or trigger the activation of the enzymes at a defined time [95] (Figure 1B). Indeed, recent studies of mycobacteriophage D29 physiology show that while holin is dispensable for phage viability, it is essential for a timely and efficient host cell lysis and phage progeny propagation [99]. LysA holin-independent lysis mechanisms are also described in some mycobacteriophages [100]. However, the *M. tuberculosis* cell envelope is a much more complex and dynamic structure compared to gram-negative and -positive pathogens, featuring a thick mycolyl-arabinogalactan-peptidoglycan complex core (mAGP) involved in virulence and persistence [101]. To be able to go past this natural barrier, most mycobacteriophages encode another enzyme located downstream of *lysA*, named Lysin B (LysB), an esterase that degrades this complex cell envelope structure by cleaving the ester bonds between the mycolic acids and the arabinogalactan [102,103]. LysB also degrades trehalose dimycolate (TDM), a glycolipid that plays an important role in mycobacterial pathogenesis [104,105] (Figure 1B). Recently, the antimicrobial activity of phage D29 LysB against *M. ulcerans* has been demonstrated in a mouse model, showing synergistic inhibitory effects with other antimycobacterial drugs in vivo and inducing an immune response by increasing the levels of pro-inflammatory cytokines IFN-γ and TNF in the DLN (draining lymph node) [106]. Besides the lysis as a primary mechanism of mycobacterial death, other authors have suggested the existence of secondary mechanisms, such as the production of superoxide radicals by lysed cells, or the induction of programmed cell death by mycobacteriophages [107] (Figure 1B). Moreover, phages can stimulate increased phagocytosis of bacteria through the opsonization of bacterial cells [108], where the phage coats the bacteria and makes it more recognizable by macrophages and other innate immune cells.

Phage-derived lytic enzymes such as endolysins are being studied as potential antimicrobial agents to combat infections caused by different gram-negative and -positive pathogens [34,109], with progresses made optimizing lysins through bioengineering [106,110,111]. Indeed, a recent study describes the novel concept of “innolysins”, an engineering approach that allows customization of endolysins by combining them with phage Receptor Binding Proteins (RBPs) to target specific bacteria [112]. Thus, “enzybiotic” is defined as a new and promising class of drugs derived from phage endolysins, characterized by a rapid mode-of-action, high specificity, and less probability to induce bacterial resistance when compared to traditional drugs [113]. Still, the use of mycobacteriophage-encoded lysins to treat TB has major challenges, including the access of these lysins to the peptidoglycan layer through the thick *M. tuberculosis* cell envelope, which gets further complicated by the fact that drug-resistant *M. tuberculosis* are thought to have even more complex cell envelope than drug-susceptible strains [15]. Thus, it is critical to determine which mycobacteriophages are able to infect and specifically kill drug-resistant *M. tuberculosis* strains. Additionally, it is critical to fully understand the mechanisms by which mycobacteriophages dismantle mycobacterial cell envelope layers in order to design new strategies using them alone, in combination, or in synergistic combination with existing drugs [106,114]. Indeed, studies indicate that gp39, a novel gene of mycobacteriophage SWU1 with unknown function, can disrupt mycobacterial lipid metabolism increasing the cell envelope permeability, ultimately potentiating the efficacy of multiple drugs against *M. smegmatis* in vitro [115].

Several efforts are being conducted to identify novel mycobacteriophage proteins with antimicrobial activity [116,117]. For instance, a small peptide derived from mycobacteriophage Che12, conserved in many mycobacteriophages, showed mycobactericidal and immunoregulatory functions in a *M. tuberculosis* mouse model [118]. Even though the use of phage-derived products has the potential to treat drug-resistant TB, there are some limitations when compared to the use of live mycobacteriophages such as these phage enzymes lack the capacity of self-replication at the site of the infection, they cannot adapt to the environment, and similar to drugs, the pathogen can adapt to select resistant clones, although it is not expected to be at a high rate [119]. Conversely, it is unknown if these enzymes are expressed in sufficient amounts by mycobacteriophages during the *M. tuberculosis*-host cell interplay.

## 3. Phage Therapy to Treat Multidrug-Resistant TB

Phage therapy was found to be successful against different pathogens such as Pseudomonas aeruginosa [120,121,122,123,124], Staphylococcus aureus [125], Klebsiella pneumoniae [126], Acinetobacter baumannii [127], Clostridium difficile [128], Enterococcus faecalis [129] or Escherichia coli [130,131], among others [21,132,133]. Furthermore, in recent years, several animal and human studies have shown promising results in the treatment of diseases caused by diverse drug-resistant bacteria [21]. This is the case of a topical mono-phage preparation successfully used to treat 6 patients with diabetic foot ulcer infected with MDR S. aureus [134]. The first clinical trial (Phase I/II) conducted using an oral phage administration against chronic otitis caused by antibiotic-resistant P. aeruginosa, also resulted in bacterial load reduction with no adverse effects reported [135]. Further, when the efficacy of phage pVp-1 was tested in a sepsis murine model to treat MDR V. parahaemolyticus, a reduction in mortality was observed using both intraperitoneal and oral routes of administration [136]. Not only phage therapy has proven to be successful against some MDR pathogens, but there is also evidence that phages might be capable of restoring drug sensitivity of such pathogens. In this regard, development of bacterial resistance to phage infection in MDR P. aeruginosa comes with an evolutionary trade-off: changes in the multidrug efflux pump outer membrane porin M (OprM), which acted as phage-binding receptor, led to decreased phage infectivity but increased sensitivity to different classes of drugs [137].

All these studies are proof-of-concept that it is feasible to use phage therapy to treat drug-resistant bacteria, including *M. tuberculosis*. So far, only a few bacteriophages have been investigated as potential therapeutic options to treat TB in different in vitro studies, including mycobacteriophages DS6A, TM4, D29, BTCU-1, SWU1 and Ms6 [reviewed in detail by [138]], with only two in vivo studies using the guinea pig model [139,140] showing promising results in the elimination of *M. tuberculosis*. However, to date, there is still limited data regarding the use of mycobacteriophages in the treatment of MDR-mycobacterial infections [37].

There are several general phage features that make mycobacteriophages suitable as therapeutic agents against drug-resistant TB [132,141]: (a) By nature, phages only infect and replicate inside their bacterial hosts without harming the human eukaryotic cells, and have the capacity to lyse and kill their hosts at the infection site [142]; (b) Their host range is mostly limited to *Mycobacterium* spp., which allows for the development of targeted therapies for specific *M. tuberculosis* strains with no apparent collateral damage to the human microbiota [143], representing one of the major advantages of phage therapy over the use of drugs; (c) Phages are the most abundant organisms in the biosphere (10^31^ viral particles estimated for the entire phage population [144,145]), isolated from multiple environments through common and well-established microbiology techniques [91], and easy to characterize using molecular biology tools [146]. New mycobacteriophages have been sequenced and characterized these past few years [73], indicating the existence of a great phage diversity [147], which could lead to the discovery of novel mechanisms to disrupt the *M. tuberculosis* cell envelope; (d) Phages are easy to engineer to contain suitable features for phage therapy; (e) Because phages have the ability to replicate inside the pathogen using the bacterial cell machinery to generate more viral particles, less number of doses are required when compared to drug treatments [143]. In this context, mathematical model for population dynamics showed that a single dose of phages is more effective than multiple doses of drugs to treat systemic infections caused by *E. coli* [148]; and lastly (f) Phages are easy to propagate at large-scale in an in vitro setting, significantly reducing the production costs when compared to drugs [132].

Nonetheless, there are still challenges to overcome before mycobacteriophages could be widely used to treat MDR-, XDR-, and XXDR-TB [149]. These are discussed below and summarized in Table 2.

### 3.1. Mycobacteriophage Host Specificity

In nature, bacteriophages present very diverse host specificities, and we can find those that infect different species within a genus or even strains from different taxonomically related genus (broad host range), whereas others only infect a particular bacterial species and/or strain (narrow host range). Mycobacteriophages belong to the latter, with their host range restricted to *Mycobacterium* and some other members of the Actinobacteria phylum [150,151]. Moreover, there are some mycobacteriophages that can only infect a particular species [152]. Most mycobacteriophages have been isolated using the non-pathogenic and fast-growing *M. smegmatis* as a host, and only a few of them have been tested and shown to also efficiently infect and lyse *M. tuberculosis* [150,151,153,154,155]. Evidence indicates a strong correlation between cluster designation, reflective of genomic variability, and host specificity [77,156]. Infectivity of *M. tuberculosis* is a feature that seems mainly restricted to cluster K and sub-clusters A2 and A3 mycobacteriophages [157], including the extensively studied TM4, L5, and D29. Members of others clusters do not infect *M. tuberculosis* or they infect at very low efficiencies. Specific infectivity of *M. tuberculosis* could be due to different factors [158] including abundance and status of the preferred bacterial host in the environment [159], bacterial host receptor nature and availability, or bacterial defense systems and phage-specific restriction/modification mechanisms, as suggested by the capacity of phage DS6A to form plaques exclusively on bacterial members of the *M. tuberculosis* complex [151]. Still, the genetic basis of bacterial host specificity remains largely unknown [156].

Mycobacteriophage specific host range could be considered as a double-edge sword in regards to potential applications in phage therapy: on one hand, phage specificity is required for successful clearance of a particular pathogen without affecting the overall human microbiome composition. On the other hand, it also represents a significant limitation, as this high specificity might create a bottleneck in terms of identifying potential phage candidates to efficiently kill *M. tuberculosis*. However the growing mycobacteriophage database [72] is proof that newly isolated mycobacteriophages are being sequenced and characterized frequently, showing a great degree of genome diversity that could potentially expand the host range possibilities of the existing phage pool [92,147]. Indeed, mycobacteriophages have the ability to switch or expand their host range frequently among different mycobacterial strains [77,156]. Such is the case of cluster G phage mutants with single residue substitutions in their tail proteins that infect *M. tuberculosis* strains more efficiently [156]. The use of directed evolution is also being explored as a tool to increase the infectivity of mycobacteriophages under different experimental conditions [160].

Because the first interaction between mycobacteriophages-*M. tuberculosis* (phage adsorption) involves the attachment of the virion onto a bacilli, one can assume that recognition of *M. tuberculosis* cell envelope surface receptors by phage tail proteins and subsequent binding will likely determine phage-bacterial host specificity and infection efficiency, although not much is known about phage binding receptors on the *M. tuberculosis* cell envelope surface [161]. Only a few mycobacteriophage targets have been described for non-tuberculous mycobacteria: mycobacteriophage D4 interacts with the glycolipid mycoside C [162]; mycobacteriophage Phlei seems to target lyxose-containing molecules [163]; and mycobacteriophage I3 probably recognizes glycopeptidolipids (GPLs) on the mycobacterial cell envelope [161]. Mycobacteriophages can also use more sophisticated two-step mechanisms, involving a reversible binding to a cell envelope saccharide, followed by an irreversible binding to a cell envelope emerging protein domain [164]. On the other hand, in the mycobacteriophage-mycobacteria recognition, several phage proteins were described. Minor tail proteins gp6 and lysin protein gp10 of phage L5 and homolog proteins in D29 seem to recognize and bind to mycobacterial surface receptors [165]. In phage Rosebush, the gp42 protein is a tail component that assists in infection through hydrolytic degradation of sugar-containing molecules on the mycobacterial cell envelope surface [156]. In order to delineate molecular mechanisms of phage-*M. tuberculosis* interactions and their specificity and modulation, phage’s tail interactions with *M. tuberculosis* bacterium attachment proteins and receptors need to be first defined. In this regard, a high-throughput approach based on transposon insertion sequencing, INSeq, was used to identify new phage receptors for multiple *E. coli* phages [166], with the potential of being used to characterize receptors in phages that infect other bacterial species, including *M. tuberculosis*. Identifying new receptors will provide insight to understand why only certain mycobacteriophage genomic types can infect members of the *M. tuberculosis* complex. In this regard, drug-resistant *M. tuberculosis* represent even a greater challenge than susceptible *M. tuberculosis*, as the former has a cell envelope composition that still remain largely unknown [15]. Recently, BSL-2 safe strains of MDR-*M. tuberculosis* that are triple auxotrophic (pantothenate, leucine and either arginine or methionine) have been developed for performing experiments in laboratories that do not have BSL-3 facilities [167]. These strains seem to be the most promising candidates to test mycobacteriophages for activity against MDR strains and also to understand their interactions.

Besides the natural ability of phages to expand their host range and to adapt to new bacterial hosts [77,156], several studies have shown that phage specificity can be altered through genetic engineering of their RBPs [158]. Thus, by using different synthetic biology strategies, one could re-program RBPs specificity and consequently modify a phage’s host range [168]. As finding phages for the *M. tuberculosis* complex and/or for a particular *M. tuberculosis* strain in the environment can prove to be time-consuming, synthetic biology approaches could represent a significant advancement in phage engineering, opening the avenue for the custom design of *M. tuberculosis*-specific mycobacteriophages for the treatment of drug-resistant TB. It would also allow to modify well-characterized lytic phages and expand its host range to efficiently infect *M. tuberculosis*. Through genetic engineering approaches one can also eliminate potential detrimental genes such as those encoding for toxins, antibiotic resistance and non-essential proteins of unknown function, as these can be transferred to the bacterial host by lysogenic conversion when a temperate phage DNA is integrated into the bacterial genome [169,170]. Because of the low lytic efficiency, the use of temperate bacteriophages for phage therapy is not recommended. Additionally, a recent study described the pathogenic role of temperate phage Pf in *P. aeruginosa* infections, wherein uptake of the virion by mammalian immune cells results in the production of phage RNA, triggering antiviral immunity through induction of type I interferon and suppressing bacterial clearance [171], which further supports the use of lytic instead of lysogenic phages.

Another key factor to consider in mycobacteriophage-*M. tuberculosis* interactions is the impact of the human lung environment in shaping the *M. tuberculosis* cell envelope. Upon infection, *M. tuberculosis* is deposited in the lung alveolar space where it comes in close contact with the lung mucosa. The lung mucosa is composed of surfactant lipids and an aqueous hypophase called alveolar lining fluid (or ALF). *M. tuberculosis* is thought to be in close contact with ALF prior to encountering host cells (e.g., AMs, neutrophils, alveolar epithelial cells) for an undetermined period of time. ALF contains a series of innate soluble molecules, including homeostatic hydrolytic enzymes (hydrolases) shown to significantly modify the bacterial cell envelope upon contact (Figure 2A). *M. tuberculosis* exposure to ALF hydrolases happens during the first steps of infection, during its escape of dying cells in the alveolar space, and in cavities during active TB. Upon 15-min contact with human ALF, two major *M. tuberculosis* cell envelope virulent factors, the mannose-capped lipoarabinomannan (ManLAM) and the trehalose dimycolate, are significantly reduced on the bacterial cell surface, among others. These ALF hydrolase-driven alterations define *M. tuberculosis*-host cell interactions in vitro and in vivo (Figure 2A), and potentially determine the infection and disease outcome [2,3,172,173,174,175]. Thus, the in vitro repertoire of potential exposed receptors on the *M. tuberculosis* cell envelope surface being recognized by mycobacteriophages might be modified during the initial stages of infection depending on the host ALF status, potentially directing phage-*M. tuberculosis* interactions in vivo. These human ALF hydrolases are also shown to be expressed at different levels and have altered functionality in elderly individuals, as well as in HIV infected and/or diabetic people compared to healthy individuals. This could potentially alter the cell envelope of *M. tuberculosis* differently, thus making defining phage-*M. tuberculosis* cell surface interactions more challenging depending on the ALF status of a given person at a given time. Moreover, during latent infection *M. tuberculosis* is thought to mainly remain intracellular within the granuloma, where its cell envelope is also thought to experience remodeling due to *M. tuberculosis* altered metabolic processes within the host cell [176,177].

Most in vitro studies use *M. tuberculosis* grown in solid or liquid medium, where bacteria have not been exposed to lung environment ALF hydrolases and thus, never subjected to the same in vivo bacterial cell surface modifications. This means that there could be well-characterized mycobacteriophages efficient in recognizing and lysing specific *M. tuberculosis* strains that have been discarded because their testing was performed using in vitro models that do not reflect the impact of the lung environment. As above mentioned, TB comorbidities such as HIV, old age, chronic lung diseases or diabetes, among others, seem to alter ALF status and functionality of different innate immune components in the lung mucosa [174,178,179], possibly driving different modifications on the *M. tuberculosis* cell envelope. As a result, mycobacteriophages that kill very efficiently a particular *M. tuberculosis* strain in a healthy adult, might not work when the same *M. tuberculosis* strain infects an elderly or an HIV co-infected individual.

We propose that mycobacteriophage receptors and mycobacteriophage-*M.tb* interactions should be well characterized and studied for each defined human population (e.g., HIV-infected individuals, elderly, diabetic, etc.) before widely implementing a particular phage therapy strategy. Definitely, the lung environment is a critical factor to take into account that requires further study when designing phage therapies to treat drug-resistant TB. How *M. tuberculosis* cell envelope changes and adapts during the course of pulmonary infection is also poorly understood, and represents a critical gap in our knowledge for defining new mycobacteriophage receptors in drug-resistant *M. tuberculosis* strains.

### 3.2. Mycobacteriophage-M. tuberculosis Interactions within Mammalian Host Cells

*M. tuberculosis* is an intracellular pathogen, known to replicate inside phagocytic and non-phagocytic cells [172,181]. In addition, *M. tuberculosis* infection of alveolar cells drives a host immune response that in 95% of the cases results in the establishment of latent *M. tuberculosis* infection (LTBI), characterized by the formation of persistent granulomas [176]. Since mycobacteriophages do not seem to have a natural ability to cross mammalian cell membranes (although some level of internalization by endothelial cells has been described for some phages [182]), the fact that *M. tuberculosis* can be found inside macrophages and/or granulomas once the infection is established, represents an added challenge for the development of TB phage therapy strategies [138]. Until recently, it was assumed that mycobacteriophage therapy would only be possible in the case of extracellular pathogens [183], but these past few years, several strategies have been developed to overcome this issue. As an example, bacteria were used as vehicles to carry lytic phages inside murine macrophages to kill intracellular methicillin-resistant *S. aureus* [184]. In the case of *M. tuberculosis*, mycobacteriophage delivery inside macrophages has been achieved using *M. smegmatis* or liposomes. Besides being fast-growing and non-virulent, *M. smegmatis* can also serve as host bacterial reservoir for mycobacteriophage proliferation to increase phage titers before reaching the targeted *M. tuberculosis* pathogen [185,186] (Figure 2B). Though *M. smegmatis* mediated mycobacteriophage delivery has been tested in vitro, due to its pathogenicity in mice models, it may not be a suitable strain to perform phage delivery studies in vivo. However, *M. smegmatis* non-pathogenic *Δesx-3* mutant strain, which has been tested as a vaccine candidate against *M. tuberculosis* in mice, could be the best carrier for delivering phages into the intracellular compartments [187]. The use of liposomes as a non-bacterial vector proved that liposome-associated TM4 phages infect mammalian cells more efficiently than free phages [188] (Figure 2B). The fact that both systems have proven successful in the delivery of mycobacteriophages inside mammalian cells, opens the possibility for further studies using these systems for the treatment of drug-resistant TB either during active TB (bacteria replicating extracellularly and intracellularly) or potentially latent TB (intracellular bacteria at different replication stages).

An alternative strategy could be to engineer mycobacteriophages to recognize and bind host cells, so these can be internalized and have access to intracellular *M. tuberculosis* (Figure 2B). This has been shown by re-engineering phage M13 to display the penton base of adenoviruses, which mediates viral attachment to integrin receptors and internalization by mammalian cells, without phage replication [189]. Pathogenic bacteria often exploit surface receptors such as mannose receptor on macrophages for phagocytosis wherein ligands such as terminal mannose, fucose or N-acetyl glucosamine are efficiently internalized [190,191]. Mycobacteriophages which are engineered in either capsid or tail proteins to carry ligands such as mannose-6-phosphate can be targeted towards intracellular compartments of macrophages. Such modification in phages is possible with the affinity tagging approaches wherein modified phages carrying affinity tags (e.g., StreptavidinTag II) can be coupled to ligands and used for phage therapy [192].

### 3.3. Bacterial Resistance to Mycobacteriophages

One of the major factors to consider in mycobacteriophage therapy is the potential development of bacterial resistance against phages over time, similar to what happens with drugs. Several mechanisms of bacterial resistance have been described so far [119,193,194,195], including altering, masking or modifying bacterial receptors to prevent phage adsorption; blocking phage DNA entry through superinfection exclusion (Sie) systems; restriction/modification systems; development of CRISPR (clustered regularly interspaced short palindromic repeats)-Cas systems; abortive infection; interference with the virion particle assembly; etc. but because phages are dynamic and constant evolving entities, they are also capable of developing counter-defense mechanisms to avoid bacterial blockade [196,197], such as mutations of RBPs to recognize alternative receptors in the bacterial surface, expansion of their bacterial host range, and development of anti-CRISPR [198], BREX (bacteriophage exclusion defense system) [199] or DISARM (defense island system associated with restriction–modification) [200] mechanisms, among others. This host-phage co-evolution can be influenced by several factors, including phage and bacteria mutation rates and associated fitness cost, population diversity or genetic variability, and it is difficult to predict which bacterial resistance and phage countermeasures will take place at any given moment [201].

In the case of *M. tuberculosis*, it should be considered that this pathogen is characterized by a low mutation rate of ~2 × 10^−10^ mutations/bp/generation [202] that, in combination with a long replication time of approximately 20 h, highly decreases the probability to develop resistance to mycobacteriophages, which favors the use of phage therapy for the treatment of drug-resistant TB when compared to other pathogens. Still, as these co-evolutionary bacterial-phage dynamics may dictate the therapeutic outcome, several strategies can be applied to shift the balance towards a sustainable phage infection reducing the risk of resistance. Indeed, the use of phage cocktails, preferred over mono-phage preparations, is one of the measures that can be applied to overcome this issue. By using a mixture of genetically diverse phages against several strains of *M. tuberculosis*, one can target synergistically a diverse range of highly conserved receptors on the *M. tuberculosis* cell envelope surface that are essential for bacterial survival and thus, less prone to mutation [193], minimizing the occurrence of cross-resistance. In addition, *M. tuberculosis* would have to develop resistance to all phages in the mix, which would probably come with a high fitness cost for the bacterium and compromise its survival within the mammalian host [137,203,204,205]. Still, it is possible that spontaneous resistant clones arise during phage treatment, so it is critical that phage cocktails are constantly tested and updated in order to target newly emerging clones [201]. For *M. tuberculosis* treatment, it has been suggested that a combination of three to six mycobacteriophages with different resistance countermeasures could be ideal [77]. Indeed, a three-phage cocktail was used in the treatment of disseminated drug-resistant *M. abscessus* subsp. *massiliense*, a non-tuberculous mycobacterium, in a 15-year-old patient with cystic fibrosis [37]. The patient was not responding to drug treatment and the clinicians, after screening the mycobacteriophage database, used genome engineering to develop a cocktail of three phages (phage Muddy, a lytic derivative phage of ZoeJ, and an expanded host range mutant from phage BP lytic derivative) to efficiently kill the drug-resistant strain. The cocktail was administered intravenously with no adverse effects and showed clinical improvement, reducing infected skin lesions and improved liver function. This study shows promising results and opens the possibility for phage therapy in drug-resistant TB.

Other strategies to reduce the appearance of phage resistance are the use of sequential treatments, in which individual phages with different characteristics are administered one after the other [206], or a combination therapy of phage and drugs [207,208,209] (Figure 2A). The latter is based on an evolutionary understanding of the fact that a higher number of simultaneous mutations are required for the bacteria to evolve and be able to overcome both phage and antibiotic selective pressures when compared to independent treatments [210]. In addition, if the pathogen develops resistance to the phage, it will probably come with a fitness cost such as decreased virulence or increased sensitivity to antibiotics (if the phage binds to a virulence factor, or if it binds to an antibiotic efflux pump, respectively) [40], which will make the bacteria more vulnerable to the drug. In this case, development of bacterial resistance to phages could be seen as a beneficial trade-off in the elimination of the pathogen. Furthermore, it has been demonstrated that phages can become more effective in killing bacteria in the presence of sub-lethal drug concentrations [208,211], a phenomenon called phage-antibiotic synergy (PAS) (Figure 2A), although some studies have suggested that sub-lethal drug concentrations can actually increase bacterial resistance and thus, high drug concentrations would be more appropriate in combination therapies [212]. Importantly, it has been shown that PAS is not influenced by the drug-resistant status of the bacteria, and that is equally valid to treat MDR pathogens [213]. Other advantages of phage-drug combined therapy are the increased and extended effectiveness, dosage reduction, and rapid clearance of the bacterial pathogen when compared to individual treatments. And so, in cases of XDR-, XXDR- or TDR-TB, when the probability of drug treatment success is really low or non-existent, using a phage-drug combined strategy could be an alternative. Finally, for specific settings and cases, the use of personalized treatments could be used to avoid the overuse of generalized phage or drug therapies that would increase the chances of developing bacterial resistance [214].

Overall, more in vitro and in vivo studies exploring the mechanisms of bacterial resistance to phages and mycobacteriophage counter-defenses are needed in order to develop successful phage therapy strategies to treat drug-resistant TB, as very little is known about those mechanisms in *M. tuberculosis* [161]. Indeed, several *M. smegmatis* and *M. tuberculosis* mutants resistant to mycobacteriophages from clusters K, G, or A3 have been isolated, but not fully characterized yet [215].

### 3.4. Mycobacteriophage Interactions with the Mammalian Immune System and the Lung Virome

Recent metagenomic studies have described the ubiquitous presence of viruses as part of the human microbiome, named “virome”, including but not restricted to the gut, urinary and vaginal tract, bloodstream, skin and the lungs [216,217,218]. Among them, phages are typically the most abundant type, found in greater numbers than eukaryotic viruses. The “phageome” has been neglected in most of the previous metagenomic studies, and it is not until recently that its potential impact on commensal microbial communities and its role in host immune responses against bacterial pathogens is starting to be uncovered. As diversification of the phageome is linked to bacterial diversity [219], phages are of special importance in modulating bacterial communities and host-bacterial interactions. Indeed, the lung microbiome is altered in cystic fibrosis patients, where mycobacteriophages and other phages are found infecting cystic fibrosis-associated bacterial pathogens [220], providing further evidence of their possible role in the clearance of pathogen infections within the human body.

The impact of the phageome in the host immune system needs careful consideration when designing treatment strategies. Ideally, a mycobacteriophage selected for therapeutic applications should act at the site of infection but not trigger a generalized immune response. Indeed, phages can interact directly or indirectly with mammalian cells, altering host innate and adaptive immune responses [108,221]. As an example of a direct interaction, phages are capable of binding glycan residues present in the mucus layer secreted by the epithelium through Ig-like domains in their capsids [222] (Figure 2A). This observed phenomenon could be of particular interest for the use of mycobacteriophages as prophylactic agents, where phages could be administered in sufficient amounts to bind to the lung mucosal surfaces, providing an extra layer of local innate immune protection against *M. tuberculosis* infections, with major implications in TB transmission. In fact, proteins with Ig-like domains have been described in several mycobacteriophages [223,224]. Whether these particular mycobacteriophages have sufficient lytic activity against *M. tuberculosis* needs further investigation. Still, this prophylactic strategy could effectively work to stop TB transmission in high burden environments, being used to protect health care professionals or household contacts. Indeed, delivery of D29 mycobacteriophages to the lungs via inhalation in a mouse TB model, as a mean to protect against *M. tuberculosis* infection, showed a decrease in *M. tuberculosis* burden in lungs at 24 h and 3 weeks post-infection, indicating a prophylactic effect where phages might be infecting and killing the extracellular bacteria before *M. tuberculosis*-host cell interactions occur [154].

More recently, it was demonstrated that phages can be translocated through epithelial cells by a non-specific transcytosis mechanism, including intestinal and lung epithelium [182] (Figure 2A), which might explain a systemic presence of phages, especially after oral administration. Systemic mycobacteriophage delivery through intravenous or oral administration may be feasible to treat TB, given that sufficient phages reach the lungs, as high bacterial burden might support phage self-replication to sufficient densities to clear *M. tuberculosis* from the infection site [225]. However, this amplification might not be enough if *M. tuberculosis* bacterial loads are low at the infection site [226], making phage treatment questionable for latent *M. tuberculosis* infection. Another potential issue with oral administration is mycobacteriophage susceptibility to the stomach acidic environment, which could be solved using microencapsulation strategies [227]. Conversely, phages delivered through intranasal or endotracheal routes could probably reach the lungs at higher numbers with limiting systemic effects, being more adequate to treat lung infections such as TB at their active stage [228,229]. Indeed, aerosol delivery of mycobacteriophage D29 was shown to be effective against *M. tuberculosis* infection and the development of pulmonary TB in the mouse model [154,155]. Other studies comparing delivery rate of D29 using different inhalation devices in different animal models concluded that a vibrating mesh nebulizer is the most efficient method for the delivery of high amounts of mycobacteriophages in vivo, whilst the soft mist inhaler might be a better choice for self-administration [230]. And so, nasal delivery of mycobacteriophages for human therapy might be a better strategy through the use of different types of nebulizers and/or inhalers such as dry-powder inhalers (DPIs) or metered-dose inhalers (MDIs) [215]. Nevertheless, mycobacteriophage stability to different environmental tissue conditions is variable, pointing out that same formulation strategies might not be applicable to all mycobacteriophages [227]. Thus, formulation of mycobacteriophage preparations for preservation and storage, effect of environmental tissue stresses on mycobacteriophage stability, release kinetics, or nanoparticle encapsulation and size (which could protect and stabilize phages against adverse environments) are some of the current questions that need answers in order to design optimal individualized mycobacteriophage preparations to treat drug-resistant TB.

It is also important to consider the reduction of phage numbers due to the activation of mammalian host innate and adaptive immune responses when determining the optimal dose/s for a particular phage treatment. Typically, phages are cleared from the human body by phagocytes [229,231] (Figure 2A), even when no phage-specific responses have been developed, although phage lambda mutants escaping innate immune responses were detected [108]. A mammalian host adaptive immune response has also been described, as shown by the production of neutralizing antibodies against phages (Figure 2A), including phages of the human phageome [232], feature that has been traditionally used to test for competence in immunocompromised individuals. In this regard, even though the number of antibodies against naturally-occurring phages does not seem to affect therapy outcome, special attention needs to be given to patients with immune disorders [233], as the mechanisms of phage-immune system interactions, as well as phage immunomodulatory roles are still unclear [232,234]. Antibody responses to phages also seem to be related to the administration route and frequency. Particularly, repeated doses of high phage concentrations might be required for successful treatment in order to reach sufficient numbers for bacterial clearance; however, multiple phage doses and/or high phage titers may over activate the immune system, leading to an increased anti-phage antibodies and clearance from the tissue [119,228] and/or adverse effects on the human host. Different strategies have been developed to reduce phage clearance, including the use of appropriate delivery systems such as liposomes to avoid recognition by the immune system [235], or the development of liquid and solid formulations for stabilizing and encapsulating phages [227,236,237,238].

Conversely, it is unlikely that phages could completely eliminate a bacterial pathogen on their own, as that would imply losing the bacterial host machinery needed for their replication and auto-preservation. While effective phage treatment could significantly reduce the targeted bacterial population, the mammalian immune system is the one that ultimately completely clears pathogen remnants from the tissue. In this process named “Immunophage Synergy”, the action of the immune system is necessary and complements the phage antimicrobial activity, as seen in neutrophil-phage cooperation [239] (Figure 2A). Moreover, phages have the potential to indirectly induce anti-inflammatory cytokines through their interaction with host immune cells, helping to reduce inflammation and tissue damage [240]. Conversely, phages also have the ability to stimulate the production of IFN-γ in a TLR9-dependent manner, linked to exacerbated inflammation in ulcerative colitis (UC) patients [241]; indicating that phage immune interactions might be case-specific and depending on multiple factors such as dose and/or host immune status, among others.

Only a few studies detailing the use of mycobacteriophages to eliminate *M. tuberculosis* complex are available, with limited data regarding the immunological effects of phage therapeutics [138,242]. Although no documented cases of anaphylaxis have been associated to phage treatment in humans, as opposed to drugs [24,243], it is critical to define the exact immunological responses in order to predict the success of new potential phage therapies to treat drug resistant-TB.

## 4. Concluding Remarks and Future Perspectives

In a “Dry Pipeline” era of drugs, where only a few new compounds are being discovered, the need for alternative strategies to treat emergent MDR, XDR, XXDR and/or TDR *M. tuberculosis* strains is a priority. The notion of phage therapy against bacterial infections has been around for almost a century due to its numerous advantages. The vast diversity observed in almost 2000 characterized mycobacteriophages opens the possibility of finding highly specific and efficient phages for the elimination of drug-resistant *M. tuberculosis* strains, otherwise associated with untreatable infections. An important concept that has been overlooked in most in vitro mycobacteriophage-bacterial host specificity studies is the impact of the lung environment in shaping the *M. tuberculosis* cell envelope. It is known that human ALF components interact with and alter the *M. tuberculosis* cell envelope, defining *M. tuberculosis*-host cell interactions, where the oxidative and inflammatory status and functionality of the ALF determine the nature of these *M. tuberculosis* cell envelope changes [2,174]. This observation could be further exploited to uncover novel mycobacteriophage receptors and to revisit the use of previously discovered mycobacteriophages that were initially dismissed and that could be highly specific when taking into account the impact of the human lung environment. This will require special attention in drug-resistant *M. tuberculosis* strains, as not much is known about their cell envelope composition, the impact of the lung environment, or the potential mycobacteriophage receptors on their surface.

Despite the obvious advantages of using phages as therapeutic agents and promising results from recent studies and clinical trials for several bacterial pathogens [244], there are still several challenges compared to traditional drugs that need to be addressed in order for phage therapy to take off and be acknowledged and widely implemented in modern medicine, being the lack of regulations one of the major obstacles, as well as the need for more robust scientific evidence [245]. Because of that, phage therapy is currently used only on a compassionate care basis. Another critical aspect is the poorly understood specific interactions of mycobacteriophages with the mammalian immune system, posing a question in the scientific community of whether phage preparations are safe for human use. Phages are known to interact with host cells and generate and modulate directly or indirectly host innate and adaptive immunity, but their exact immunomodulatory properties and specific mechanisms are still largely unknown. Ideally, a mycobacteriophage used in phage therapy should reach the lungs in sufficient numbers and be able to work synergistically with the host innate immune system (and drugs, in a combined phage-drug therapy) to clear the infection, but without causing a dysregulated systemic immune reaction or adverse effects. Special attention needs to be given to the excessive generation of phage-neutralizing antibodies, which can represent a hurdle in phage therapy applications by rapidly clearing the phages before eliminating *M. tuberculosis* [40]. Thus, a perfect balance in the activation of the host immune system needs to be reached for a successful outcome, which will likely depend on the administration route, the dose, and the host’s immune status, among other factors. In this regard, a concept to explore would be the use of phages to generate a low-level protective immune response to prevent *M. tuberculosis* infections. TB comorbidities such as aging, HIV co-infection, and chronic disease such as diabetes alter the host’s immune response [174] and consequently, a particular phage treatment that would work for a healthy individual might not work in these cases, stressing the need for more personalized treatments.

The fact that phages are evolving entities is considered an advantage for phage therapy, although it can also constitute a limitation. On one hand, it can help phages overcome the development of bacterial resistance and also reduce the treatment dose thanks to their ability to replicate in the presence of their host, but at the same time, phages can acquire harmful properties by horizontal gene transfer through the interaction with the human microbiota. Thus, there are also important evolutionary consequences related to phage therapy that need to be addressed before it can be implemented, such as the effect of phages in complex bacterial populations or phage-bacterial host co-evolution. To start addressing this, methods are being developed for a more controlled phage therapy by conjugating phages to gold nanorods, whose excitation by near-infrared light causes localized heating that destroys phage-infected bacteria, without harming nearby mammalian host cells [246].

Currently, due to mycobacteriophage therapy clinical practicality, this may be only used on the most difficult or virtually incurable cases of drug-resistant TB, where drug therapy does not work or is insufficient. Phage therapy alone may never completely replace the use of traditional drugs, but combined mycobacteriophage-drug therapies working together and enhancing the stimulation of the host immune response might prove to be the most efficient in the long run for several reasons. First, it makes sense from an evolutionary point of view, as it can prevent the potential appearance of bacterial resistance to mycobacteriophages, as previously discussed. Second, it has been suggested that both phages and drugs in combination are more effective in controlling pathogenic bacteria than either alone. This synergistic effect has been observed in numerous studies, independent of the drug-resistance status of the pathogen [208], which, in the case of TB, would allow shortening of treatment duration and/or reduction in drug dosage. Consequently, this could prevent further development of drug-resistance. As a direct benefit, the standardized use of combined therapies in MDR patients worldwide could reduce the conversion, and thus, cases, of XDR and XXDR-TB, as these are generated from MDR cases failing their treatment. To what extent mycobacteriophages in combination with drugs can be used as routine therapy is difficult to assess, especially in TB endemic areas that are linked to low-income regions. Using mycobacteriophages as prophylactic instead of therapeutic agents might be a better strategy in these regions, where therapy-associated costs and patient non-adherence to treatment represents a huge problem and potentiates the emergence of drug-resistant *M. tuberculosis* strains, further increasing the treatment costs. By eliminating *M. tuberculosis* before the potential establishment of the infection, transmission is drastically reduced, decreasing treatment costs. In this regard, regular mycobacteriophage administration in high-risk individuals (e.g., household contacts of TB patients) could be exploited as a prophylactic measure to prevent onset of TB disease instead of preventive TB therapy, which uses drugs. This strategy would generate non-mammalian host-derived immune protection in the lung mucosal surfaces by using highly specific and effective phages that could quickly kill extracellular *M. tuberculosis* before the establishment of the infection, or work together with the host innate response in this matter. Still, the impact of the constant presence of mycobacteriophages in the lung microbiome and the host immunity generated by them would need to be well-established prior to even considering their therapeutic use.

These potential strategies might be used in the near future, where exploiting the use of mycobacteriophages to aid resolving drug-resistant TB cases could be a valid option when drugs fail to do so. However, before this, selective, effective, safe, and standardized phage preparations will need to be established [247], that in combination with drugs and the host immune response, might help to treat otherwise untreatable TB. In this scenario, the development of bioinformatics pipelines to predict phage and *M. tuberculosis* binding sites, phage efficacy, etc., that account for the host tissue environment, will be required to define phages with highly specific recognition and killing efficacy against drug-resistant *M. tuberculosis* strains.

## Figures and Tables

**Figure 1 ijms-22-00735-f001:**
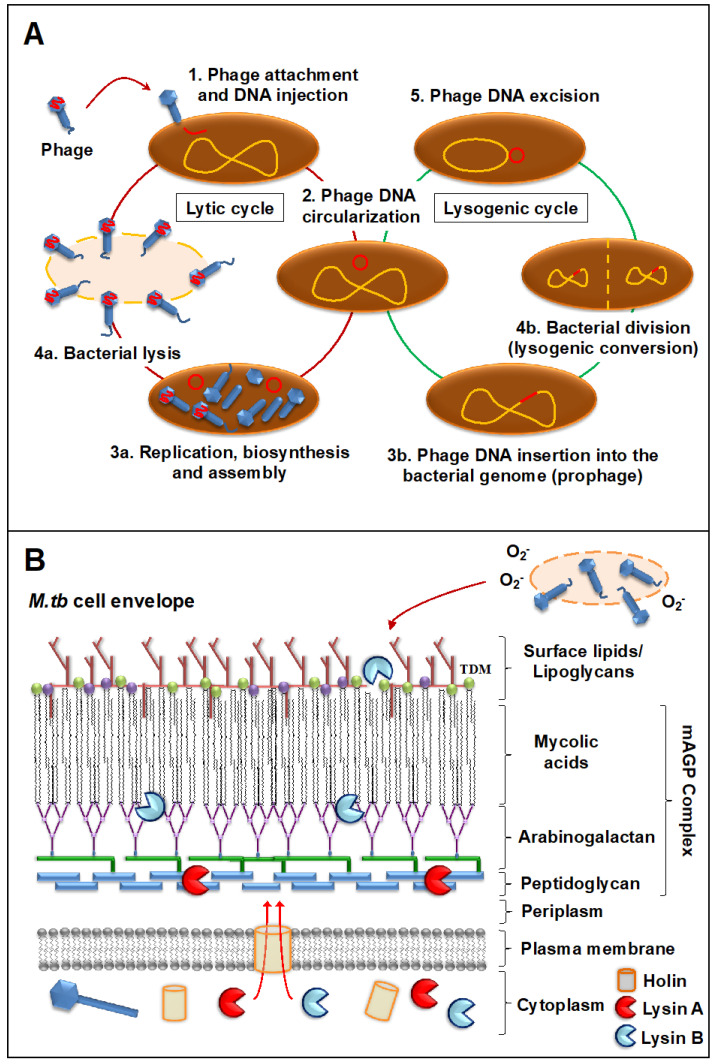
Infection of *M. tuberculosis* by mycobacteriophages. (**A**) Steps during mycobacteriophage infection: (1) The phage attaches to *M. tuberculosis* through specific receptors and injects its DNA (red); (2) Phage DNA circularizes inside the *M. tuberculosis* bacillus; then, specific environmental cues will determine if the phage undergoes a lytic or a lysogenic cycle. If the lytic cycle is induced: (3a) New phage DNA and viral proteins are synthesized and assembled into new viral particles and; (4a) Viral particles will be released after the lysis of the *M. tuberculosis* bacillus. If the lysogenic cycle is induced: (3b) The phage genome is integrated into the *M. tuberculosis* chromosome, becoming a prophage; (4b) The prophage will replicate along the *M. tuberculosis* genome and will be transmitted to the progeny that will acquire new properties encoded in the prophage (lysogenic conversion); (6) Under certain triggers, the prophage DNA will be excisioned from the bacterial chromosome and the lytic cycle induced (3a–4a). (**B**) Most mycobacteriophages rely on endolysin-holin systems to kill their hosts. Holins act as membrane proteins to help translocate the lysins to reach their targets: Lysin A degrades the peptidoglycan layer, whereas Lysin B cleaves the ester bonds between mycolic acids and the arabinogalactan in the mycolyl-arabinogalactan-peptidoglycan (mAGP) complex, disrupting the *M. tuberculosis* cell wall core (mAGP). Lysin B is also known to degrade trehalose dimycolates (TDMs) in the outer layer. Although lysis is the primary mycobacteriophage mechanism for bacterial death, secondary mechanisms such as the release of superoxide (O_2_^−^) radicals from lysed bacilli might also contribute in the elimination of *M. tuberculosis*.

**Figure 2 ijms-22-00735-f002:**
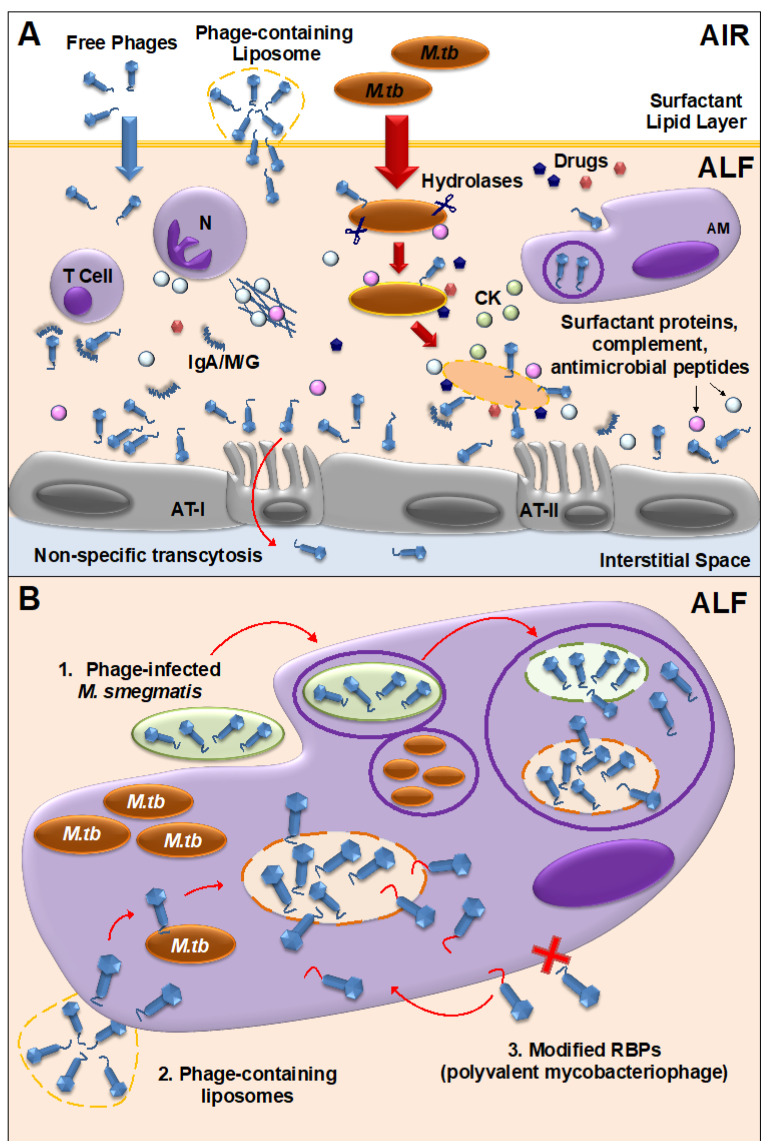
Mycobacteriophage-*M. tuberculosis* interactions in the human lung alveolar environment. (**A**) Free and/or encapsulated mycobacteriophages (e.g., in liposomes) reach the lung alveolar space, cross the surfactant lipid layer and are bathed in alveolar lining fluid (ALF), where they can directly or indirectly interact with innate immune cells, generating a local immune response. Some phages will be cleared by phagocytic cells [e.g., alveolar macrophages (AMs)], or by neutralizing antibodies (IgA/M/G). If delivered in sufficient numbers, mycobacteriophages are capable of binding glycan residues in the mucus layer secreted by alveolar epithelial cells type II (AT-II) through Ig-like domains in their capsids, providing an extra non-host antimicrobial protection layer against *M. tuberculosis* infection. Mycobacteriophages can also be translocated to the interstitial space through the epithelium via a non-specific transcytosis mechanism. When *M. tuberculosis* enters the alveolar space, innate soluble molecules in ALF, including hydrolases (represented as scissors), modify the *M. tuberculosis* cell envelope, driving *M. tuberculosis*-host cells interactions. These ALF hydrolases-drive alterations will define *de novo* exposed *M. tuberculosis* surface receptors to be recognized by mycobacteriophages. Here is depicted a combined mycobacteriophage-drug therapy, where drugs and mycobacteriophages work together with an activated mammalian immune response [including neutrophil (N) degranulation and neutrophil extracellular traps [180] secretion, AMs activation, and cytokine and chemokine secretion] to kill *M. tuberculosis* and control the infection. (**B**) Different strategies can be used to deliver mycobacteriophages inside mammalian cells (e.g., AMs) to access intracellular *M. tuberculosis*: (1) Mycobacteriophage-infected *M. smegmatis* bacilli acting as vehicles are phagocytized by AMs being delivered into *M. tuberculosis*-containing phagosomes. Replicating mycobacteriophages within *M. smegmatis* lyse the bacterium accessing the phagosome lumen where they recognize, infect and subsequently lyse and kill *M. tuberculosis*; (2) Liposome-associated mycobacteriophages, which infect mammalian host cells more efficiently than free phages; (3) Generation of polyvalent mycobacteriophages through receptor binding protein (RBP) engineering to recognize mammalian immune cells in addition to receptors on the *M. tuberculosis* cell envelope surface. Note: This figure depicts a general overview of the mycobacteriophage-*M.tb* interactions in the human lung environment and with the mammalian immune system; and thus, does not fully represent the complexity of the processes described in the text.

**Table 1 ijms-22-00735-t001:** Applications of mycobacteriophages in TB.

Applications	Notes
Typing of clinical *Mycobacterium* isolates	• Use of pre-defined host-specific mycobacteriophage sets to identify new *M. tuberculosis* isolates.
TB diagnostics	• Use of reporter phages (e.g., LRP, Φ^2^DRMs, etc) to determine the presence of viable bacilli (active disease), *M. tuberculosis* drug susceptibility, and TB treatment efficacy (e.g., identifying presence of persister bacilli).
Mycobacterial genomic tools	Use for mycobacterial genetic manipulation including: gene exchange, allelic exchange, recombination and transduction of chromosomal markers, point mutations, transposons and insertions to generate *M. tuberculosis* mutants. Use to introduce immunity-based selection markers in mycobacterial hosts (immunity to superinfection).
Prophylaxis and therapeutics	Use of whole phages or mycobacteriophage-derived lytic enzymes to efficiently lyse and eliminate mycobacterial hosts. Use as combined synergistic therapy: mycobacteriophages/endolysins + anti-TB drugs.

**Table 2 ijms-22-00735-t002:** Phage therapy challenges in drug-resistant TB treatment.

Challenges and Limitations	Potential Solutions
Host specificity	Global phage database screening. Phage’s host range expansion using directed phage evolution and/or bioengineering. Development of screening bioinformatics tools to identify targeted *M.tb* host virulent factor epitopes (e.g., efflux pump).
Unknown impact of human ALF on the *M.tb* cell envelope	• Identify how the *M.tb* cell envelope adapts (changes) to the different environments that encounters at different stages of infection [e.g., contact with ALF, within the phagosome, extracellular, within granulomas or cavities, or when being transmitted (in sputum)].
Phages access to intracellular *M.tb*	Novel phage delivery systems [e.g., *M. smegmatis* (Trojan horse concept), phage microencapsulation]. Phage bioengineering to recognize well-defined macrophage receptors (the mannose receptor or MR).
*M.tb* resistance to phages	Use of different phage cocktails. Phage-drug combined treatment (phage-drug synergy) in combination with the mammalian host immune response. Phage sequential treatment. Phage personalized treatment.
Overactivation of the mammalian host immune system and risk of anaphylaxis	Optimize phage delivery routes. Establish phage dosage and frequency. Maximize synergy between phages and the mammalian host immune system.
Lack of phage therapy regulations	• Standardize global regulations for phage production (under GMP conditions).
Phage cytotoxicity to the human host	Use of highly lytic phages that do not integrate into the *M.tb* genome. Targeted phage genetic bioengineering to remove potential phage virulent factors to the mammalian host. Define function of unknown phage genes.

## Data Availability

No new data were created or analyzed in this study. Data sharing is not applicable to this article.

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
