# Peer review of "Mycobacteriophages as Potential Therapeutic Agents against Drug-Resistant Tuberculosis"

_ijms, 2021, doi:10.3390/ijms22020735_

Round 1

Reviewer 1 Report

The authors cast a valuable insight in the world of mycobacteriophages based upon their own expertise and as retrieved from extensive literature study. Although the road to actual therapeutic usage of mycobacteriophage is still far off. More so because M.tb is much better adapted to withstand the host immunity than the phages preying on them. However, the authors have an honest approach dealing with the subject, so I have no objections to their opinion.The review will be quite valuable for scientists new to the field of mycobacteriophages or researchers that are developping tools based on this technology. The manuscript is also well written and reads quite easily. The created images by the authors delivers a complete picture of the written review and adds significantly to the quality.

In summary, a great review for a specialized domain.

All the best,

Davie

Author Response

We thank the reviewer for the favorable comments, time and effort dedicated to read and evaluate our manuscript. We really appreciate it.

Reviewer 2 Report

The review prepared by Allue-Guardia et al., is interesting and contains a lot of relevant information about mycobacteriophages that can be used as potential therapeutic agents against drug-resistant tuberculosis. However, there are some specific points which have to be improved:

  1. In the chapter Introduction (lines 83-87), the Authors give some information about the experimental phage therapy. Please provide more details.
  2. In the chapter Mycobacteriophages (lines 104-126), the Authors describe the mycobacteriophage applications in M. tuberculosis. Please create the table or scheme that summarizes this information.
  3. In the chapter Mycobacteriophages (lines 135-149), the Authors describe the architecture of the genomes of mycobacteriophages, Please create scheme that summarizes this information.
  4. In the chapter Mycobacteriophages (lines 149-154), the Authors describe the clusters and sub-clusters of mycobacteriophages. Please add some details.
  5. The figure 2 is too simply and unreadable. Please improve it.

Author Response

We thank the reviewer for the time and effort spent in evaluating our manuscript, and for all the comments and suggestions, which have helped to improve the manuscript. Please, find below in red the responses to the specific comments. Changes in the main manuscript text have been highlighted in grey.

Comment: English language and style are fine/minor spell check required. 

Response: We thank the reviewer for this comment. We have double checked our grammar and a native English speaker read and provided suggestions to improve the quality of the written language. This have been incorporated along the manuscript without changing the original meaning and content of this review. 

Comment 1: In the chapter Introduction (lines 83-87), the Authors give some information about the experimental phage therapy. Please provide more details.

Response: As requested, we have added additional information regarding the use of experimental phage therapy as approved by the FDA (lines 86-95). In particular, experimental phage therapy was used in a clinical setting involving the treatment of a complicated case of Mycobacterium abscessus infection. Examples of phage therapy cases and clinical trials to treat M.tb and other pathogens are provided in section 3 - Phage therapy to treat multidrug-resistant TB.

Comment 2: In the chapter Mycobacteriophages (lines 104-126), the Authors describe the mycobacteriophage applications in M. tuberculosis. Please create the table or scheme that summarizes this information.

Response: As suggested, we have expanded the mycobacteriophage applications paragraph and added new references (lines 112-116; 122-125; 139-140), and included a new table (Table 1) summarizing this information. 

Comment 3: In the chapter Mycobacteriophages (lines 135-149), the Authors describe the architecture of the genomes of mycobacteriophages. Please create scheme that summarizes this information.

Response: Although mycobacteriophages have a general shared architecture, these are structural mosaics; thus, we consider that a figure of the overall mycobacteriophage genome architecture may be too simplistic to represent the current genome variability, and might distract from the focus of our review -  Phage therapy to treat multi-drug resistant TB. Instead, we have provided references of previously published studies containing detailed descriptions and figures of the mycobacteriophages genome and variability (references 77-90).

Comment 4: In the chapter Mycobacteriophages (lines 149-154), the Authors describe the clusters and sub-clusters of mycobacteriophages. Please add some details.

Response: We thank the reviewer for this comment. We have included more details regarding clusters and sub-clusters in mycobacteriophages (lines 165-178), which we hope will provide a better overview of this section. A more detailed explanation on how the classification system was created and clusters/sub-clusters allocated, is published elsewhere and has been referenced in the text (e.g. Hatfull 2018).

Comment 5: The figure 2 is too simply and unreadable. Please improve it.

Response: We agree with the reviewer that Figure 2 is simple. This was done in purpose, as we tried to summarize the complexity of these interactions for the general reader. As mentioned by Reviewer 1, the created figures deliver a complete picture of the written text and adds significance to the quality of this review. We feel that complicating this figure will detract from the goal of this review that is to introduce the complex world of the mycobacteriophages to young investigators entering the field. Said this, we made fonts and the figure bigger and easier to read, as well as improved the figure legend emphasizing the complexity of the processes (lines 441-444), but keeping it simple in the figure to allow the general reader to assimilate easier the discussed concepts.

Round 2

Reviewer 2 Report

Thank you very much for improving the manuscript and taking into account my suggestions. In this form, the manuscript may be published in the IJMS journal.